# Walking in the City: Christian Spirituality in Amsterdam through the Eyes of Michel de Certeau

Erica Meijers 

Protestant Theological University, 9712 HA Groningen, The Netherlands; p.e.m.meijers@pthu.nl

**Abstract:** This contribution investigates the spiritual position of a Christian congregation in urban contexts of gentrification and de-churching. A project in Amsterdam will serve as a case to explore crucial issues for shrinking congregations in 'up-and-coming' neighbourhoods, who aim to transcend the insider/outsider dichotomy between the congregation and its (urban) context. The project at hand shows a shift from exclusively Christian congregations to communities of people with various outlooks of life and from professional structures to cooperation between professionals and volunteers. Using the work of the French theologian Michel de Certeau as a city guide, and his understanding of the empty tomb as a key theological concept, the paper reflects on epistemological and methodological questions brought about by these shifts. After that, issues closely connected to the observed shifts are discussed: the questions of language (how to deal with different ways to express and interpret experiences) and ownership (who is in control in situations of plurality). The article argues for an urban Christian spirituality based on an epistemology of not-knowing and otherness, informed by methodologies of receptivity and desire, leading to practices of multilingualism and open ownership-structures.

**Keywords:** community development; urban theology; gentrification; de-churching; Michel de Certeau; exposure; otherness

## 1. A Church in a Gentrifying Neighbourhood

How to be a Christian community in a city marked by gentrification and de-churching? This is, very shortly, the question which the protestant congregation of the Nassau church in Amsterdam is struggling with. On Sundays, a group of about forty people gather to celebrate in the monumental church; the rest of the week, both members of the congregation and other neighbours walk in and out the building for all kinds of cultural and social activities. The congregation cannot afford a full minister anymore and is therefore increasingly run by the church members themselves. Dependency on temporary professional help (for instance for pastoral care) and incidental professional services (worshipping and preaching) weaken the continuity within congregation, but increasing initiatives from volunteers strengthen the ownership of the community. The congregation continues to shrink and is aging, but thanks to its long-time commitment to the neighbourhood, it keeps many relations to other groups and projects. It considers itself to be part of the larger community living in the neighbourhood and aims to contribute to its wellbeing: "We are looking for unexpected connections which increase humanity and solidarity in the city. We want to investigate which desires live among the people in the neighbourhood and among ourselves", [. . .] because "God can be known through the other".[1]

In this article, I will investigate questions related to the search of this urban congregation for God "through the other". It is the old question on Christian spirituality in today's society, which every generation is confronted with anew. In a situation in which churches are of little importance to the majority of inhabitants (in 2019, 15 percent of the population of Amsterdam was Christian, and this number is decreasing[2]), roughly three reactions can be observed according to the Dutch practical theologian Hendriks (1999): first, this situation is

ignored and the church carries on as it always did; second, the problem of survival becomes the focus of all ecclesial activities; and third, the church lives its Christian spirituality as an active engagement in modern society. In 1995, Hendriks described the Nassau church as a community of the last category.[3] Thirty years later, however, the congregation is confronted with new challenges. The church is situated in a neighbourhood close to the city centre which has changed in about twenty years from a poor and rebellious neighbourhood to an upcoming neighbourhood with a mixed population of middle-class newcomers (families and young urban professionals) and older inhabitants belonging to the working class and precariat (often single). A third group of inhabitants came to the neighbourhood during the squatter's period in the 1970s and 1980s and is now mostly part of the middleclass. They are still very socially active and engage to bridge gaps between the poor and the wealthy. The congregation has members from all three groups; its connections in the neighbourhood are mostly with the second and third group.

Part of the social housing in the area has been sold. As a consequence, the prices of living went up, the population changed (families are leaving the area), and small family businesses disappear one after the other. The interaction between local politics, demographic developments and economic change (from an industrial to a service-and capital-oriented city) has changed Amsterdam to a city for the well-educated (higher) middleclass (Boterman and van Gent 2023). The lower middleclass, however, is leaving the city (Jansen 2023). These developments weaken the social tissue of the neighbourhood, which is of greater importance to the old inhabitants than to the new, who engage less with their direct environment (Dorssers and Meijers 2021, p. 50; Jansen 2023). This process, known as gentrification, is fuelled by the process of capitalisation: the dynamics of money and real estate slowly transform people into consumers and commodities. Those who do not generate money are useless and overlooked (Sassen 2014; Meijers 2015a).

As one of its responses to these developments, in 2021, members from the congregation and the independent organization *Church & Neighbourhood* organized an interactive exhibit in the church. Local residents were invited to share their stories: What memories did they have about the neighbourhood? Did they (still) feel at home? What did they want to change? The aim was to make space for the stories of those who are overlooked, to build connections between groups in the neighbourhood ("look at the neighbourhood through the eyes of others", said the leaflet) and to find new ways of belonging to the neighbourhood as a church. It is this project, of which I was part myself as both a member of the congregation and a researcher[4], which has brought up the issues I want to address in this article.

The project *A(t) home in the Staats* (the name of the neighbourhood is *Staatsliedenbuurt*; the streets are named after 19th century statesmen) was a collaborative project in two ways: the organizers were church members and other inhabitants of the neighbourhood, and of professionals (social workers, academics, artists) and volunteers. This is characteristic for a situation more and more current in Western European urban contexts: congregations are too small to act alone, have no or only little professional theological guidance, but feel part of and want to engage in society. The shrinking budgets cause a shift from professional to voluntary work in and by the church. The shrinking congregation fuels the need for cooperation with others.

There are many practical questions concerning leadership, decision-making, finances and organization with regard to this situation. However, from a theological perspective, other questions seem to be more urgent: what is the theological basis of these collaborations between Christian members of the congregation and others? How does the cooperation between (theological and other) professionals and volunteers challenge our ecclesial identity? How do we express and interpret experiences and how do we deal with differences? In this contribution, I concentrate on epistemological and methodological questions (Sections 3 and 4), and in relation to these, on issues of language and ownership (Section 5). After that, I will come back to the question of Christian spirituality in changing cities (Section 6). In Section 2 I explain why I draw on the work of Michel de Certeau, while at

the same time giving some background to the theological tradition of the congregation at hand.

## 2. Bricoleurs and Poachers: De Certeau and the Nassau Church

Our guide in this article is the French historian and Jesuit theologian Michel De Certeau (1925–1986). His work is promising for urban theology for several reasons. Despite the fact that De Certeau was a Jesuit and the Nassau church is a reformed congregation, they both draw from ecumenical traditions and are looking for new ways for church and theology. De Certeau contributed to the theological thinking after Vatican 2. He was deeply touched by late modern secular society and tried to develop Christian spirituality as an active engagement in a world in which God has become a hidden mystery. As a priest and intellectual living outside his order, he explored ways of believing in the city, developing a material and spatial methodology from below. He presented and investigated experiences which usually do not appear in classical ethnographic research methods like participant observation and interviewing. Recent methods, like, for instance, research with 'go-alongs', is partly inspired by the work of De Certeau (Kusenbach 2003, p. 19). De Certeau was interested in small details which reveal particular properties of urban places. Thus, he followed traces of desires for another life, maybe of the hidden God. Religious experience in modern, multi-layered urban contexts was defined by De Certeau as something that is always elsewhere. God, too, is always somebody or something else, always coming towards us. His confession of faith: "Without you, I cannot live. I do not hold you, but I hold on to you. You are always other and I need you, since the deepest truth about myself is between us" (De Certeau 1987, p. 8).[5] Many of De Certeau's theological texts may look like practical or systematic-theological theories, but they might be better understood as reflections on this prayer. Among historians and urbanists, his work is well known, but theologians have only started to give his work more attention after the turn of the millennium (Bauer et al. 2019, p. 9; Theobald 2016, p. 3). His eloquent poetic stumbling on the opaque experiences of modern everyday life makes De Certeau an interesting companion for an urban theological journey. He practices what Heather Walton wants practical theologians to be: "practical mystics and mystics of practice", or, "poets of the broken form", "bricoleurs"—a term she borrows from Walter Benjamin and Michel de Certeau. The importance of this approach, to Walton, is to "recognize and respond to [...] what [and who] is damaged, derelict and yet possessed of piercing power". (Walton 2019, p. 15). To De Certeau, this 'other' power is difficult to detect because it is covered by discourses of people in power. However, it is in these hidden places of otherness, that a spiritually strengthening power can be found. It is here that the mystery of God in modern city life might be hidden. Theopoetics are not only expressed in words; they give room to other ways to articulate knowledge. That is why this paper is accompanied by images of artist Philippe Velez McIntyre (Figures 1–7).[6]

Like De Certeau, the congregation of the Nassau church sees itself as part of the mission of Christian engagement to the margins of urban daily life as a place where God is present. The first years of this engagement (1979–1995) were described by one of the ministers of this period, Paula Irik. She was influenced by theologians like Karl Barth, Oepke Noordmans, Heiko Miskotte and Hans Hoekendijk. She describes the fundament of the urban mission of the congregation as the belief that Gods light is to be found in the depth of daily hardship and yearning. God cannot be kept within the boundaries of church life and dogmas, she says, but God is in the world, because of his overflooding grace and love (Irik 1995, pp. 118–19). Following Noordmans, she looks at creation (in concreto: the neighbourhood) from the perspective of recreation, never the other way around[7].

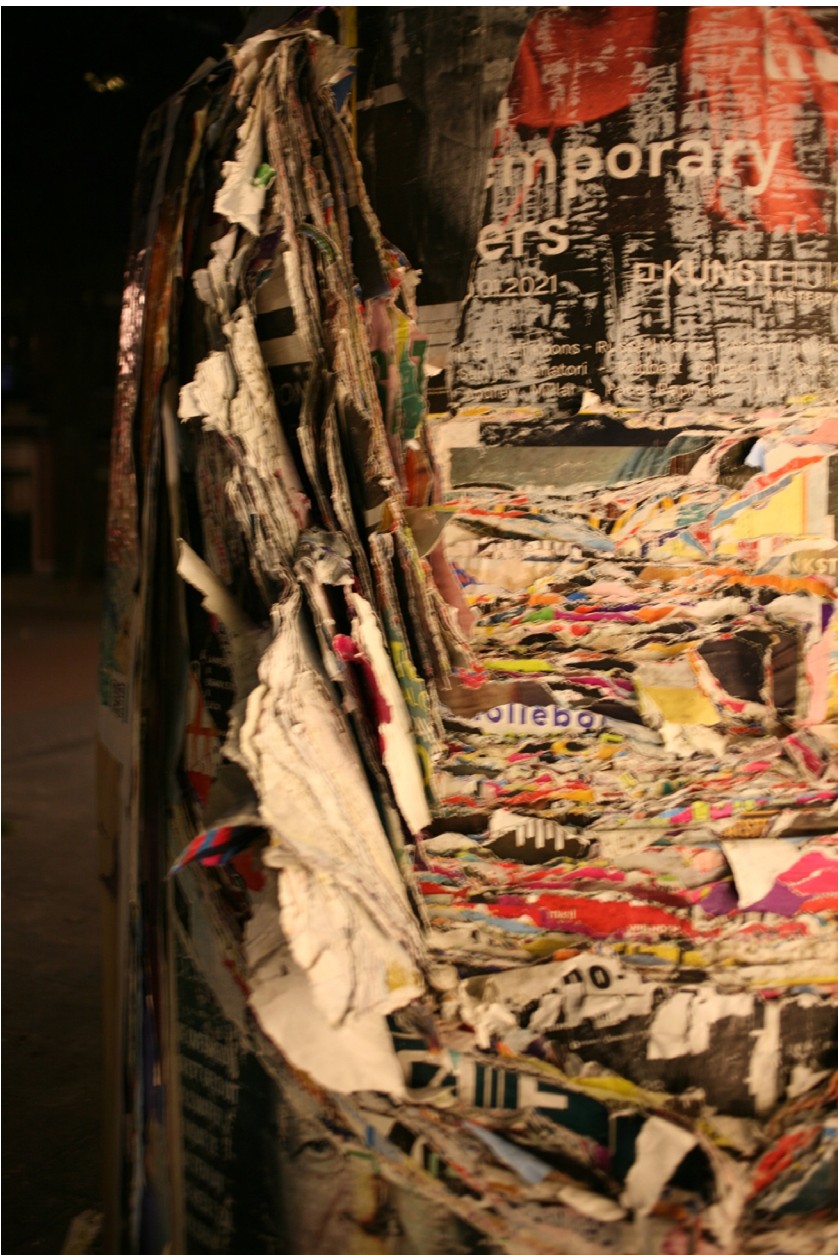

**Figure 1.** Credits: Philippe Velez McIntyre 2021.

Hoekendijk states that the Church has forsaken the city in the decisive moment in the 19th century when it built churches for the wealthy, reserved seats for them and neglected those suffering under urban violence caused by industrialization (Hoekendijk 1966, p. 114). Steps of the church into the secular city therefore can only be humble.

This is in line with the 'theology of apostolate', developed during the Second World War and confirmed by the Netherlands Reformed Church in the synodal document Christen-zijn in de Nederlandse samenleving: Herderlijk schrijven vanwege de generale synode der Nederlandse hervormde kerk (1955): modern society has lost its innocence and pride after two devastating wars. The uncertainty of modern men and women is shared by the Christians among them. Therefore, collaborations between Christians and "people of good will" for the wellbeing of society are possible and necessary. "The community of Christ is defenceless in its way through the world. (Christen-zijn in de Nederlandse samenleving: Herderlijk schrijven vanwege de generale synode der Nederlandse hervormde kerk 1955, p. 46). This position was closely related to a changed theological attitude towards the



authority of the Church, the confessions and Scriptures, and towards other religions and the State. Miskotte, one of the authors of the synodal document, proposed as early as 1949 that the Church should turn its face to the World, where God's Kingdom is present. The Church is not to work for its own survival but be the Herald of the Kingdom. The relation to the State and to secular society is now directed towards justice and peace, not toward a Christian nation (Meijers 2008, pp. 129–31).

De Certeau shares this perspective: he sees no special position for Christians in the secular world: they are as distraught as anyone else. He even takes it a step further than Christen-zijn in de Nederlandse samenleving: Herderlijk schrijven vanwege de generale synode der Nederlandse hervormde kerk (1955) by comparing the role of the Church to the position of the stranger in society (De Certeau 1987, p. 114ff). The Church should not just be with modern humans in general, but more specifically with the outsiders within that modern society. De Certeau does not belong to the so-called 'theologians of secularity" like Cox (1965), who aimed at developing a positive attitude to secularization. He has a very critical attitude towards modern society. Spiritual practices are therefore always related to the search for a political praxis. For instance, he rejected the tendency in church and theology to concentrate on 'meaning-making' as a (non-conscient) reaction to the disappearance of the search for God in society. According to him, this reduces theological questions to something outside material reality. Political, social and economic differences are overlooked and not taken seriously. De Certeau even speaks of a "new kind of heresy", which understands reconciliation as a simple plurality without confronting the tensions (De Certeau 1969, pp. 91–92). During his travels through South America in the 1960s, De Certeau was influenced by the theology of liberation (Bauer, p. 28). In Europe, he maintained close relations to the movement of working priests. Their theology of presence influenced him as well. He understood their engagement as a way of "letting outsiders into the Christian systems of interpretation" (De Certeau 1969, p. 72). Understanding Christian spirituality as a journey of desire in situations of alienation and difference, De Certeau was interested in the micro-resistance of daily tactics against the strategies of systems that control and discipline, be it those of the state, the church or of capitalistic forces. In urban spaces, he noted, "daily life creates itself through thousand ways of poaching" (De Certeau 1980, p. xxxvi).

God's option for the poor has inspired the Nassau church to engage in initiatives in the neighbourhood for and with those who are "not known by others anymore, who lost their self-respect, who are socially and economically discarded", but who certainly have their own ways to survive and claim their dignity, Irik writes (Irik 1995, p. 13).[8] "In God's love for the downtrodden his love for every human becomes visible" (Irik 1995, p. 123). She is constantly at the lookout for the joy of liberation, which she finds in a community trying to create spaces for experiment and difference, a community she defines as 'a motley crew', referring to the congregation including those without formal membership (Irik 1995, p. 121). In this then disadvantaged neighbourhood in Amsterdam, one of the first initiatives was restaurant Filah (Persian for wellbeing), a place where you could and still can eat for very little money. A thrift store was set up and a place to walk-in, have a coffee and get some advice. Slowly, a community of bricoleurs ('the motley crew') came into being, changing and struggling, adding their experiences and poaching practices to the palimpsest of stories and lost steps that constitutes a neighbourhood. After Paula Irik left the congregation in 1993, she worked as a pastor with cleaners in offices and factories, continuing her journey of theological presence.

De Certeau's poetic-political engagement with the daily spaces of urban life makes him a fascinating guide in today's streets of Amsterdam. His works helps to look more closely and formulate more accurately. Moreover, as a theologian of otherness, De Certeau might shed some light on a possible urban spirituality of Christians, on their relations to others, their language and ways of ownership and finally on the desire of the congregation to meet God through the other.

### 3. The Empty Tomb

De Certeau's work is based on an epistemology of defencelessness, or not-knowing. The modern experience of God as the one we cannot define nor possess, nor keep behind doors of institutions, rhymes with his perception of the empty tomb. According to him, the question initiating Christian beliefs (in plural), is the question of the desperate Maria of Magdala in the story of John 20: "Where are you?" All knowledge and spiritual authority end in the face of the absence of Jesus' body. Therefore, the empty tomb represents a non-place, an otherness that cannot be eliminated, but to which all Christian wisdom refers. This rupture, De Certeau says, has installed Christianity. The absence of Jesus causes a lack of reality and truth which cannot be (ful)filled, and opens up the possibility to follow and interpret Jesus in many different ways. Thus, his body is present in this plurality of 'Christian' beliefs and practices, but none of these can ever repeat or bring back what happened during the days of the gospel (De Certeau 1987, pp. 209–2013). This is the meaning of De Certeau's prayer: "Without you, I cannot live. I do not hold you, but I hold on to you" (De Certeau 1987, p. 8) (Figure 2).[9]

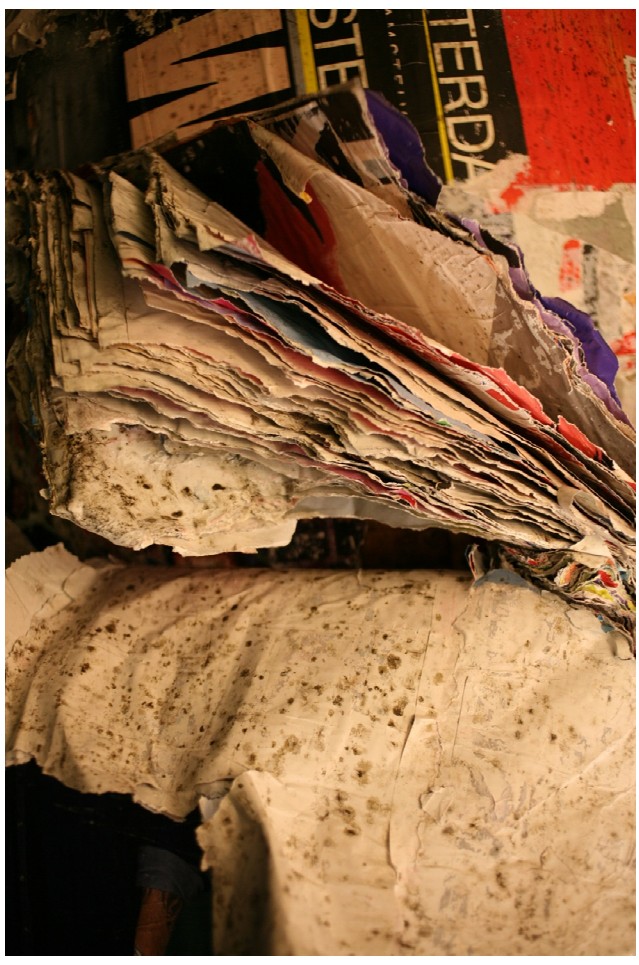

**Figure 2.** Credits: Philippe Velez McIntyre 2023.

Epistemologically, the not-knowing itself becomes a source of belief. The fundamental otherness of Christ as the Risen One sends us out to the otherness of our fellow human beings, in a ceaseless yearning for the Resurrected, whose presence remains hidden in multiple experiences and beliefs. Until Christ's Kingdom comes, the only source of theological knowledge lies between people, between people and their spaces and places and between people and God. This source is always moving and changing, since our relations to scriptures and its secret of the empty tomb differ in every context and every era. Thus,

the Roman-Catholic priest De Certeau democratizes faith: Christianity is the ever-changing relation to the absence of Christ and the constant desire for his new presence. Until then, all loci theologici refer to a truth that they do not own (Spallek 2019, p. 377). To De Certeau, the task to reform and redefine this relation, or, as he says, "to take the risk of the present" (De Certeau 1987, p. 76) is at all times a communal task. We have to look for God at places where questions about humanity are asked. The encounter with God is impossible without the encounters with others, both from the past and at present. Relations thus become a locus theologicus themselves. Within these fluid communities, "every particular witness is indispensable for the collective experience of infinity, and must in the same time accept the experience of others as necessary" (De Certeau 1969, p. 10).

The epistemology of the empty tomb implies that there is no specific Christian experience or position in today's world, only a specific Christian desire, born out of the gap between the absence of the Resurrected and the promise of his coming. The gap itself creates the possibility to believe, implying that spiritually, we cannot own anything and should be suspicious of all forms of control, since they might tempt us to close the open tomb or to fill its emptiness with powers that deny its life-giving power. Language needs constant reinvention, deeds constant reformation. God is neither in our (religious) language nor in our (good) deeds, but God might encounter us in our longing for His presence, as happened to the travellers to Emmaus (De Certeau 1969, p. 79). God is always the Coming One.

The central words in the fragile theological universe of De Certeau—absence, relations (communities) and desire—open up spaces for alliances between churches and many others in the city. But practicing these words requires a specific gaze.

**4. The Eye of the Walker**

Ever since its first steps into the neighbourhood in the 1980s, the congregation of the Nassau church has practiced what we call 'exposure'. This is part of their urban mission as described above. Exposure means opening up to urban reality, receiving not sending. This results in new relations which give rise to new (common) steps. Exposure was developed in deprived urban neighbourhoods after the second World War, drawing from theologies of liberation, black consciousness and presence. In the Netherlands, it was started in Rotterdam by Kor Schippers, Herman IJzerman and Mpho Ntoane and developed by many others since (Van der Spek 2010, chp. 1; Van Waarde 2017, chps. 2 and 3; Meijers 2017, p. 127). Internationally, it is often referred to as the cable-approach (community-based learning) (Porkka and Pentakäinen 2013) or considered to be part of community development (Haugen et al. 2022). According to De Certeau, who can be regarded as one of the founding fathers of this type of urban mission, exposure is a work of longing, not of taking possession. The moment one opens up to another space or another person, a journey begins. Without it, there is no way forward. (De Certeau 1969, pp. 5–6). The epistemology of not knowing is fundamental to this approach. It is a way of searching God, based on the beliefs that first, God is already present in the city but is hidden from our eyes (De Certeau 1969, p. 79; see also pp. 82–83) and second, that daily city life is not a field of application of theological knowledge but a source of belief which comes to life through the relations with others (De Certeau, in Bauer and Sorace, p. 39, footnote 176). De Certeau shows that the method (or rather, approach, since it can never be repeated in the same way) of exposure is deeply spiritual.

Usually, exposure is practiced by professional ecclesial or diaconal workers, but from 2016 on, the Nassau church started to do exposure as a congregation. Before, the workers spent time in the streets, sat on banks in the park, talked to representatives of other organizations in the neighbourhood, hung around until new relations grew and from there, together with others, built up new activities. One of the results of this work in the 1980s was the now independent organization Church & Neighbourhood, in which the activities mentioned above were brought together. But as the neighbourhood became more and more up and coming, and the congregation shrinking, it felt a new round of exposure was



needed. Some members of the congregation, known as the 'exposure group', organized exposure walks, this time by members (volunteers) of the congregation (I am a member of this group and took part in the activities). We were supported by the *Kor Schippers Institute*, specialized in "experience-based learning" as a form of urban mission.[10] This brought new momentum to the congregation; the Church council looked at the exposure group as a group opening up new possibilities and leading the way forward.[11]

We met on Saturdays in the church, spent a few hours in the neighbourhood and came back to share stories. Sometimes we started out with whoever was interested, sometimes we invited specific groups of the congregation, like deacons or members of the liturgical group. We tried to include as many people as we could, activating different ways of believing and practicing. We became researchers of what was going on, we discovered in our own neighbourhood what we had not noticed before: the small facade gardens through which people appropriate small spaces in front of their house, the messy backyards which makes you wonder who lives there and what is happening there, the new (higher) prices in a well-known café on the square, the overwhelming twittering of sparrows on a wall, a playing child in a red coat, dancing and clapping people around a bride in traditional Turkish cloths in front of a church centre, tourists who ask you to take their picture on the 'chicken-bridge', the shield 'Red Dawn' on a neighbourhood centre that once had been a communist stronghold. In short: traces of the past and signs of today's life. The walkers brought back their observations, coloured by their own desires and biographies. The roaming provoked feelings of connectedness and curiosity, it raised the awareness of how much remains hidden, but also confronted the walkers with feelings of fatigue and loneliness. Memories were awakened, for instance by a bridge named after Robiënna, a little girl that had been murdered years ago causing distress among the children of the neighbourhood.[12] There is no way to generalize these experiences, nothing is abstract, everything is particular and specific and related to all of our senses. Since building new relations was the goal, our being affected did not hinder us but, on the contrary, was something we were deliberately looking for: to build relations, you need to be involved personally.

The mix of traces and signals of the socially turbulent past of the neighbourhood and of new realities of young urban professionals, was confusing and familiar at the same time (Figure 3). The question of Maria of Magdala, Jesus, where are you? stood as a life-sized question mark in the middle of the cluttered street life. It evoked many other questions: 'who are we?' and 'who else is here?' Do we still belong to this city? As a congregation, as local inhabitants? Who feels at home and who is excluded? These questions gave rise to the project *(A)t home in the Staats*. We started by interviewing ten neighbours about their ways of belonging to the neighbourhood and had their portraits taken in their favourite places. These were displayed at the church, along with a huge collage of the past and future of church and neighbourhood. The interviews and photos were published in a one-off magazine. We also organized guided history tours through the neighbourhood, writing and painting workshops and evenings of storytelling and political debate (Meijers 2023).

To De Certeau, the central aspect of exposure is to get yourself into the messiness of the streets and give up on a "bird's eye view". It means to behave like an inhabitant, not a city planner. De Certeau calls for the end of "the fiction" of the planner, researcher or any other professional who is used to looking down as a god, creating distance. According to him, this position only knows "cadavers" (De Certeau [1980] 1990, pp. 140–41).[13] He proposes a view from below where visibility ends. In order to meet the 'elsewhere', where the closeness of the living God is present, one has to adopt the view of the walkers, "whose bodies follow the thicks and thins of the urban 'text' they write without being able to read it" (De Certeau [1980] 1990, p. 141). The researcher is just one of these walking practitioners making use of spaces that cannot be seen; her/his knowledge of them is "as blind as that of lovers in each other's arms". She is too close to gain an overview, but close enough to get touched. Thus, "they [the walkers] write a manifold story that has neither author nor spectator, shaped out of fragments of trajectories and alterations of spaces", which

remain "indefinitely other". (De Certeau [1980] 1990, pp. 141–42). This is, according to De Certeau, the only way to prevent the mystery, the secret of life from being killed. As both a member of the congregation, an inhabitant of the neighbourhood and a theological-diaconal researcher, I participated in the search of the congregation for new relations, and I am now giving my insights and experiences without any claim to offer the only interpretation. This text has been read and commented on by others connected to the Nassau church and by participants in the exhibit *(A)t home in the Staats*.[14] The project was an attempt to articulate what the walkers see, hear and feel and to listen to their talk about this place they call their home. For the congregation, it was part of its search for God in the neighbourhood and its wish to contribute to shalom. The non-Christian partners in this project used other words: they wanted to strengthen the social tissue of the neighbourhood and contribute to its wellbeing.

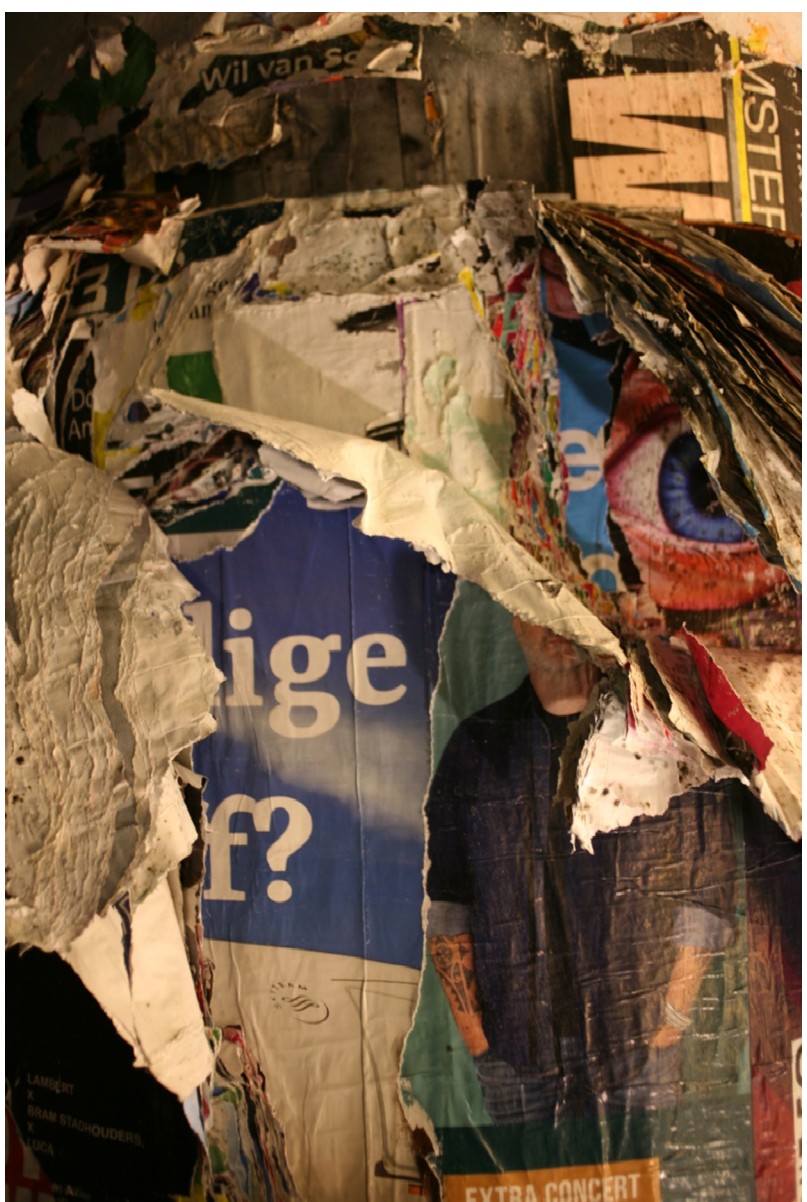

**Figure 3.** Credits: Philippe Velez McIntyre 2023.

## 5. The Chorus of Idle Footsteps

How important is this difference in language? Languages are not 'just' different ways of speaking of the same things but represent different ways of relating to reality. Above,

from a theological perspective, I have proposed epistemological and methodological foundations for the cooperation between Christians and others, professionals and volunteers. Now, I turn to two issues we encountered in the practice of our project *A(t) home in the Staats*: the issue of language (how to deal with different ways to express and interpret experiences) and ownership (who is in control in situations of plurality).

### 5.1. The Language of Trolley Cases and Postcards

De Certeau makes an intimate connection between language and place. "To live somewhere, is to create stories. To stir up or restore this narrativity, is a task of making a place habitable again" (De Certeau et al. 1994, p. 203). This involves engaging in the city as a "theatre of a war on stories" (De Certeau 1980, p. 203). The clash between religious and secular languages is only one the conflicts that can occur. In the daily practice of our project *A(t) home in the Staats*, possible conflicts between Christian and other interpretations of city life did not take place (at least not openly), because other conflicts seemed more relevant to our common wish to create spaces in which stories could meet. The clash between powerful discourses from above and the stories of the streets prevailed. Publicity posters talking the language of consumption and sign-boards of building companies or real estate agents speaking of gentrification and demographic change stood against less audible street-talk: gossip in the grocery store, parents in the schoolyard, etc. There are many layers of communication in urban areas. We concentrated on collecting hidden stories motivated by our own wonderment: who was now living in the former rental place that was renovated and sold? What was the small-talk at the flower shop on a Saturday morning about? What were the beer drinkers in the park chatting about, claiming one of the picnic tables as theirs? These stories are like the little obstinate paths of those who do not walk around the fancy grass triangle with iron edges a city planner has built in the middle of the street, but instead go straight through it, creating little sand paths of trampled grass. To De Certeau, paths like these are micro-subversions, tactics of inhabitants to counter the big economic and political strategies they cannot change, to regain some sort of ownership. De Certeau calls this "poaching" (De Certeau 1980, p. xxxvi). Today we might refer to it as 'hacking' (Meijers 2015b).[15] Worldviews that are not visible on posters and signboards resist "the privatization of stories", in which all otherness, all particularity and opacity disappear in an anonymous and general story told by city marketeers, managers and others in charge of the city, who hope to attract tourists and investors. (cf. De Certeau [1980] 1990, p. 162) In order to make encounters between different residents possible, in our project, we hoped to create space for those who do not have the power to make themselves heard or to "articulate the knowledge that has become silent" (De Certeau [1980] 1990, pp. 162–63) (Figure 4).

How did this chorus of idle footsteps (De Certeau [1980] 1990, p. 147) sound? A few quotes from participants, pointing at the changes in the neighbourhood through gentrification and growing tourism: "the sound of trolley cases of people using this house as AirBnB is disturbing, just as the change of shops: the old pet shop of which the owner knew everybody, is replaced by some pizza hut". The opposite of this is an experience by the same person: "Once I was walking in the last days of my pregnancy past Kock the coalman. He sat there, chatting away with another guy, commenting on everything around them. I stumbled past them with my big belly. "Do you send a postcard next week?" they called out to me. I like this social connection very much" (Dorssers and Meijers 2021, pp. 46–47). When the popular night shop in the neighbourhood closed down after 54 years, unable to find somebody to take over the shop (partly due to regulations concerning the retail business), neighbours offered a glossy with stories to the owner. They all expressed the same message, formulated very adequately by one of the contributors: "Late at night, tired of dancing and a bit hungry, I entered the store for the nice nuts only for sale here, a bread roll or the cat food I had forgotten to buy; a little bit of everything and always something to read and see in the hallway with the art projects and posters of activities in the neighbourhood. 25 years later, I still come to the shop, now with my daughter. This is

a store with real nice stuff, there is space for every human being (staff who might not be able to work elsewhere, are here in their element). Clients are welcome the way they are, without any judgement of how they look. Here is liberty and happiness, not anonymous, but engaged" (Jansen et al. 2023, p. 41). To the Nassau church, the evening market was an important place of reference: the building is hidden between other buildings, but there was nothing easier than to say: the church is just opposite the night shop!

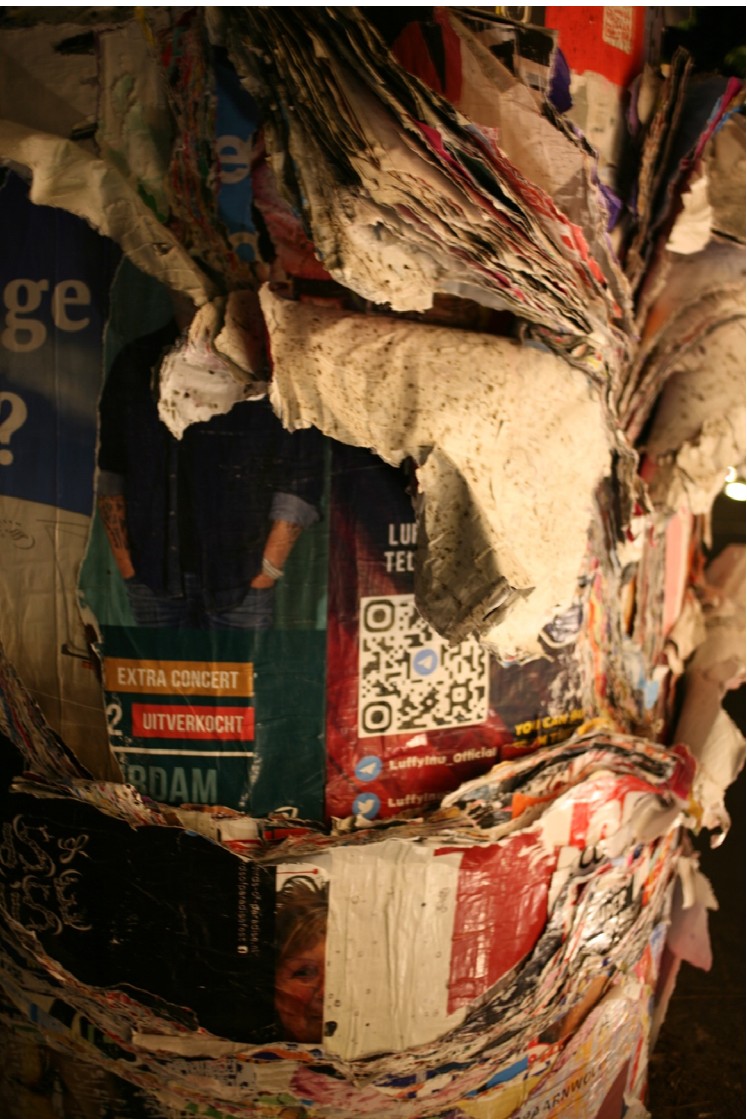

**Figure 4.** Credits: Philippe Velez McIntyre 2023.

Another woman commented on the housing situation: "I came here as a squatter. I lived in Glasgow, Aberdeen, London and Paris, but I am not a world citizen. This neighbourhood is my village, the next area is a foreign place. But now, there are a lot of people here I do not know anymore. I used to talk to everybody. Yesterday I found a note on my door with a telephone number. It said: 'I want to live here'. The free and wild days are gone now, it is all so much more commercial and anonymous. I used to be part of the community, now I am just a poor indebted person sitting on the Rotterdammer bridge, talking to all the people passing and hanging out with the homeless" (Dorssers and Meijers 2021, p. 20). A Muslim woman who founded an organization for women in the neighbourhood shared the following story: "We always meet with a group of women in the park, walking towards the cemetery. What a beautiful place. Under the overpass we start screaming. We use it as a therapy, because we are often together with people who

are having psychological problems. Just scream! I always remind myself that people with difficult behaviour have their own experiences. Even this person who was beating women wearing a head scarf. We went to the police and talked about it, but I know this guy has his own story" (Dorssers and Meijers 2021, p. 27).

How to articulate the life and desires of these particular stories and do justice to the differences that also exist between them? How to capture what De Certeau calls "the movement [...] produced by masses that make some parts of the city disappear and exaggerate others, distorting it, fragmenting it, and diverting it from its immobile order" (De Certeau [1980] 1990, p. 154)? People are put in motion by the remaining relics of meaning, and sometimes by their waste products, the inverted reminders of great ambitions. Things that amount to nothing, or almost nothing, symbolize and orient walkers' steps: names that have ceased precisely to be "proper" (De Certeau [1980] 1990, p. 158).

Except storytelling, *A(t)home in the Staats* also used visual language. Collages were made of the history of the neighbourhood and the engagement of the church in the neighbourhood, participants were invited to paint and produce their own images, which were shown in the exhibit. The first image the organizing group embraced was a picture of a trash container symbolizing the ongoing processes of renovation in the neighbourhood and thus the effects of gentrification. The waste in the picture brought up the question of what and who disappears in the process of upgrading. It focussed our attention on what is thrown away, considered useless for 'progress'. A city developed from above, organized in a functional way, cannot but produce waste, says De Certeau: "there is a rejection of everything that is not capable of being dealt with in this way and so constitutes the "waste products" of a functionalist administration (abnormality, deviance, illness, death, etc.). To be sure, progress allows an increasing number of these waste products to be reintroduced into administrative circuits and transforms even deficiencies (in health, security, etc.) into ways of making the networks of order denser. But in reality, it repeatedly produces effects contrary to those at which it aims: the profit system generates a loss which, in the multiple forms of wretchedness and poverty outside the system and of waste inside it, constantly turns production into 'expenditure'" (De Certeau [1980] 1990, p. 144).

By turning an eye to this 'waste', we wanted to make it visible, (re)value its 'piercing power' (Walton 2019) and honour the particularity of each story and image. De Certeau speaks of an 'anti-museum' to explain how stories of a place are bricolages (makeshift things), composed with the world's debris (De Certeau [1980] 1990, p. 161). They are being tied to lost stories and opaque acts, and brought together in intuitive collages. They create new realities, which cannot be fixed or organized (Figure 5).

De Certeau does not use Christian language to create a difference between Christians and others; the gospel and Christian tradition sharpens his look and helps him to interpret what he sees in the city. The most important concern of De Certeau seems to be not to kill what is alive and not to fix what is always moving. It is there that his theological concern becomes visible. The mystery of life (God) cannot be put in a museum (concept, system, unchangeable words). Dogma must be understood as doxa (glory): stammered praise of the mystery that moves us. De Certeau uses his fine language skills to open up new spaces for understanding, not to pin down or appropriate fragile (hidden) stories with (Christian or other) language and thus take away what is life-giving: the particularities of people's own language. If we cannot 'read' God into the stories of others, does this mean that there is a contradiction between the desire to find and respect otherness and the search for God in these spaces of otherness? Do we appropriate as soon as we use Christian language? To answer this question, we have to remind ourselves of the empty tomb as a non-place, representing otherness that cannot be eliminated. Thus, we can never point directly to stories, places and events as dwelling places of God. When we think we might have found such a place, we find ourselves—again—staring in the empty tomb, confronted with otherness. We can only refer to this otherness, bringing it to the table where we remember the ministry of Jesus, who valued people that were considered worthless and who created new desires and hope where there was despair and suffering. Jesus'

ministry qualifies the character of the otherness of the empty tomb and thus direct our gaze. No religious language can identify the otherness of other human beings with the otherness of God. Even when it comes to Jesus, the empty tomb reminds us that we entrust ourselves to a mystery by confessing that he is the Messiah. A mystery which leads our lives and our walks through modern cities. Our references to the otherness of the empty tomb and its quality of caring for justice and the wellbeing of others, are the fabric with which our spirituality is woven. It is a spirituality which, blind as a lover, walks towards the Coming One.

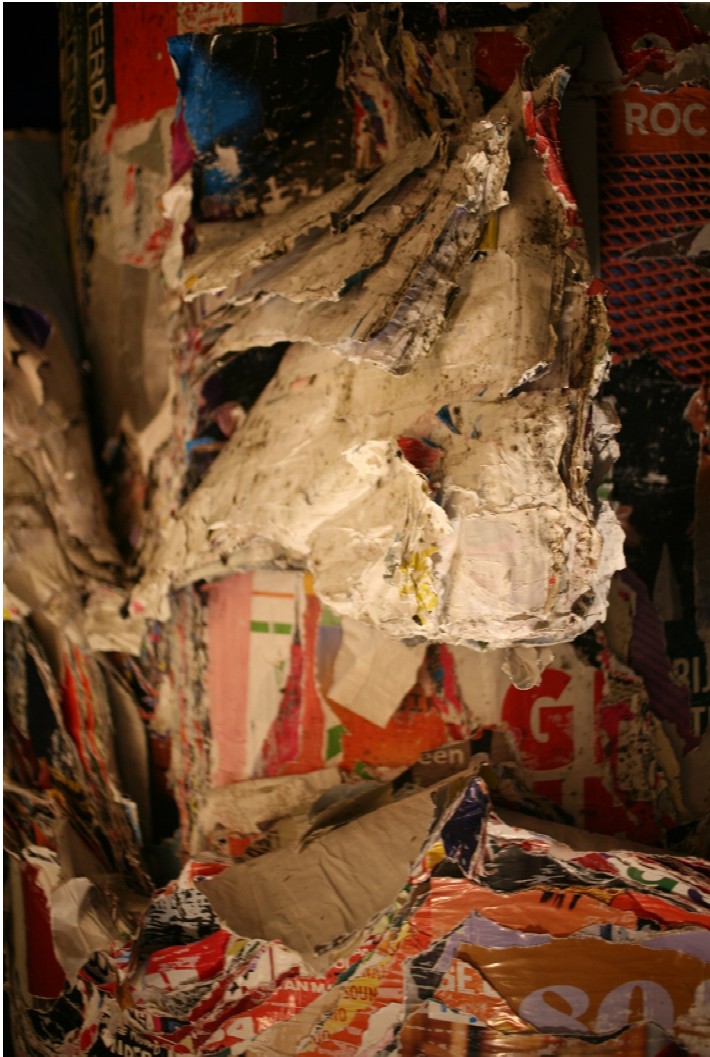

**Figure 5.** Credits: Philippe Velez McIntyre 2023.

During this walk, in order to be relevant to the reality of the city, theological and spiritual language is valuable, as long as it makes connections with different types of speech (other disciplines, but also bureaucratic and political language and multiple slangs of the neighbourhood) and as long as it engages in the "theatre of a war on stories" instead of retreating into the realm of 'meaning-making'. We refer to stories and memories of the Christian tradition to approach the mystery, whereas others use other references. To take account of the fact that we can only be fundamentally multilingual as Christians in today's society, all theological language needs to be constantly challenged by other languages to stay alive itself. It needs to open up spaces in which there is enough room for the otherness of the experiences and language of the other.



*5.2. Ownership*

Who is in control in situations of plurality, like projects of Christians and others, and professionals and volunteers? In other words: whose was the project *(T)huis in the Staats*? The open spaces in terms of language also apply to the question of ownership.

Formally, the exhibit was organized by the congregation and the independent organization *Church & Neighbourhood*, but there was no corresponding hierarchical structure. We did not have the time to sit and wait for board meetings and the official boards did not had the street-knowledge needed to take decisions and did not want to control the process.[16] Neither the professionals nor the board member in our group were officially in charge. Thus, as a mixed group, with relations to several institutions and traditions, we made our own rules and tried our own ways of decision-making. The professionals profited from their skills and networks, but the volunteers did the same. We made no difference in this respect, except that those financially depending on their contributions were given money first when we received funding. The structures were messy, which was not always pleasant. Nevertheless, it was exactly this lack of formal structures which helped participants to take ownership. There was nobody to ask for permission or complain to if things did not work out. People took initiatives, mobilized their networks and did what their hands found to do. A year later, we noted somewhat surprised that, most probably because of this open and messy structure, the new (and renewed) connections made during the exhibit, had proven to be sustainable (Meijers 2023, p. 8)

In order to reach our goal to strengthen relations in the neighbourhood, the question of ownership had to be left open. Or better: we needed to have a multiple, shared ownership with open spaces for new initiatives and languages. Collaborations between professionals and volunteers change the situation of authority: there is a shift from a formal authority to an authority based on the mission of the community and a shift from strictly organized spaces for decision-making to open spaces in which the other can talk, act and take (provisional) ownership. Or, as De Certeau formulates: there is no Christian fundament for authority, because its fundament is absent. The tomb was empty. What remains, are relations (in plural!). They can only point to a truth they do not own. (Spallek 2019, p. 367; De Certeau 1987, p. 78 ff).

What we learned from this experience is not that formal structures are to be rejected. For long-time commitments, formalizing decision-making structures can be helpful to ensure spaces for others to step in, especially in situations where multiple sources, perspectives and motivations play a role. Christian spirituality, as an active engagement in urban society, requires a type of structure which 1. seriously reckons with difference and alterity; 2. is fundamentally relational, based on equality and respect for each other's sources and experiences; and 3. builds open spaces to expand ownership to others who are not 'in' yet.

All three require an attitude of trust instead of control. Authority without fundament can only be effective by relating to the mission of the community instead of referring to ecclesial institutions. The theology of apostolate already pointed out that the relation of the Church to the State and secular society should be directed towards justice and peace, not towards a Christian nation, the Church or to Christianisation (Meijers 2008, p. 131). But the situation has radicalized since the 1950s. Today, we need to honour the fact that urban ecclesial communities are imbedded in multiple relations larger than their baptized and confirmed members; we need to (re)design church buildings and decision-making structures to enhance a larger ownership of buildings and projects, and we need to rethink the financial principles and the meaning of the (professional and voluntary) offices. However, this article does not intend to elaborate practical details of these type of structures but to reflect on the spirituality carrying them. I therefore return to the question of the engagement of the Christian Community in urban life.

## 6. Urban Christian Spirituality

How to be a Christian community in a city marked by gentrification and de-churching? After having explored the context and the challenges involved, we go back to this question

and to the three options Jan Hendriks proposed. The first is to carry on as always. This comes with the risk that the language, structures and methodologies of the late modern Western society affect the Church, without us really reflecting on it. The second option is to focus on the survival of the church by treasuring traditional theological language and ecclesial structures of ownership, regardless of the fact that not many of our partners are familiar with them, let alone appreciate them. The risk is twofold: either we isolate ourselves (and become less and less relevant in this world) or we keep on 'sending' instead of receiving, implying a Christian superiority. The third option (the choice of the Nassau church): we live our Christian spirituality as an active engagement in modern society.

This article has shown that this option is not easy to live. The experience of a shrinking congregation has nothing romantic, it confronts us with a lack of resources, people (both professionals and volunteers), skills, knowledge, etc. And when we accept it as our situation, it confronts us with the otherness of the others and inevitably with ourselves. A critical approach to our own position, traditions and behaviour is unavoidable. Just like the particular experiences of spatial otherness, which we encountered during our exposure, the otherness of other people is not an abstract issue either; it is expressed in differences in power, in social position and income, in colour, gender and sexuality, in culture and religion, etc. In relation to the neighbourhood, we have to reflect on our position. Even though Christians are now a minority in the city, they still live in a culture formed by their history and tradition, and many still keep memories (and some dreams) of the dominance of Christianity. Moreover, many of the networks, financial properties and structures of this dominance are still in place. We cannot live a Christian spirituality today without recognizing these memories and relating to these traces of dominance. In addition, the struggle to express our longing for God in the midst of urban experiences is not only a struggle with differences between Christians and others but also within Church and theology. The Nassau church is not a community of people who express their beliefs in the same way, and many conflicts have occurred within the congregation. We have learned by not avoiding emotions, by listening to each other and by realizing that none of us can appropriate Gods word. We also studied. One of the books read in the congregation was written by the Dutch theologian Theo Witvliet, on Christian identity in a pluralist Western society (Figure 6).

Witvliet proposes an open space that recalls the empty tomb of De Certeau. His proposition to a practice of transgressing borders presupposes that we do not deny the very existence of borders. Differences are real. To make real encounters possible (those that change both partners), Witvliet proposes to respect "the empty centre", an image he borrows from the French political thinker Claude Lefort. The empty centre is the space of God. Even though open spaces in daily life are always already taken by some sort of (political, economic, cultural and/or religious) power, the metaphor reminds us to let ourselves be touched by others without appropriating them. Witvliet even suggests that this open space makes it possible to receive non-Christians into the sacred and hospitable space of the Eucharist. Christians and others together can form a circle to share the bread and wine, around an empty centre, which is God's eschatological space. (Witvliet 2003, pp. 172–73). Theologically, this might evoke many issues, but in practice, it already happens in many small urban congregations, including the Nassau church: we share our lives with others and we therefore invite them into our most intimate space, a little bit in the same way as the working priests, by going to live with them, invited factory workers in to the circle of those interpreting Christianity. Our community is larger than our membership. Instead of isolating our most sacred space and keeping it from others, we admit that it is not 'our' space, but it is Christ who invites everybody to His Table. As we commemorate the death of Jesus, we form a circle around the mystery of the empty tomb. We do not own it; we only refer to it because we are touched by it. Everybody is invited to relate to Christ and it is not up to us to decide who can or cannot. Like Maria Magdalena, we are taken by desire for encounters with the Spirit: Jesus, where are you? The space which, formally, makes the borders between Christians and others visible, the Table of the Lord, is also the place

where our solidarity with all humans who long and suffer for human justice and wellbeing is most tangible. Here, we are confronted with our own vulnerability. The Eucharist calls us not to look away from the wounds in the body of Jesus. Like the empty tomb, the Table of the Lord, too, is a place where we cannot stay; it is a space of transgression, a condensed moment of receiving and giving within the ongoing journey through the palimpsest of urban life. The movement of exposure to the streets of our neighbourhood, where we receive and then share, originates at this Table (Meijers 2019, pp. 88, 95).

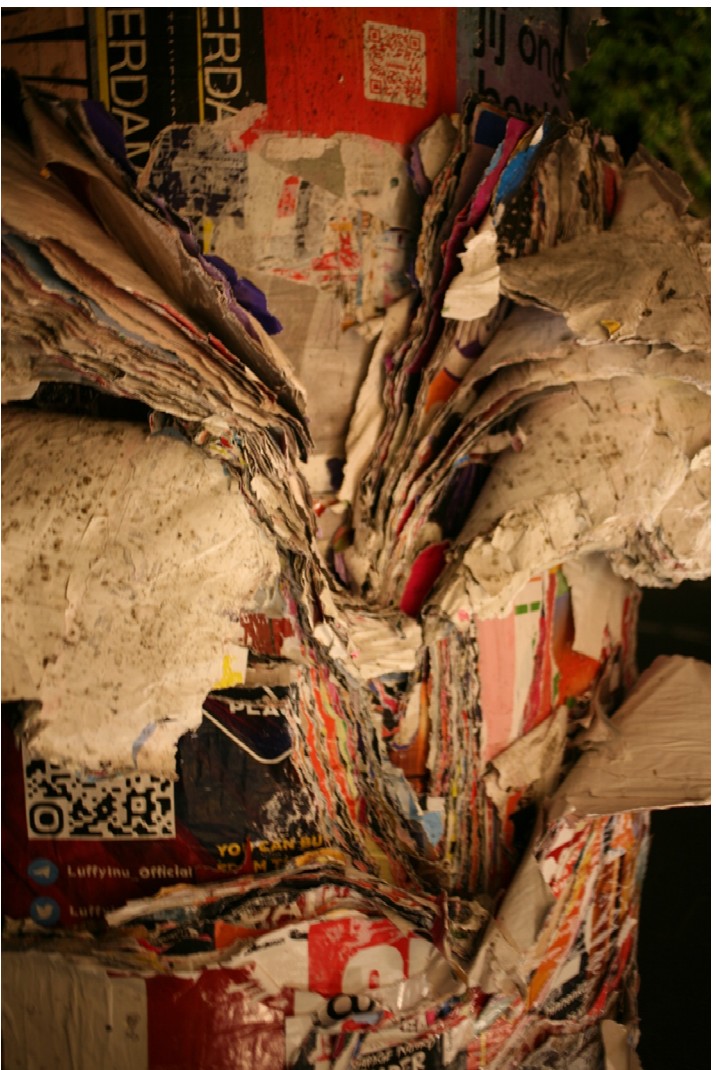

**Figure 6.** Credits: Philippe Velez McIntyre 2023.

The desire for the Coming One, if we take it seriously as a desire (not more, not less), demands an epistemology of not-knowing and methodologies of receptiveness, leading to multilingualism and open forms of ownership, reflecting the absence of Jesus' body in the empty tomb. Our faith roots in the shock that Christ is not there, in the realm of the dead. So he must be under the living. His generous Spirit can be encountered in places we do not expect. Our experiences of God are situated in otherness, in spaces outside of our own control. We cannot hold Him, but we can hold on to Him.

## 7. The Silence in-between

In his ground-breaking essay on the concept of history, Walter Benjamin wrote that time is not empty and homogenous, but "every second of time is the strait gate through which the Messiah might enter" (Benjamin 1940, p. 261). One could say that Michel

de Certeau translated this perception of time into an awareness of space and place. He approaches space as a field of desire, sometimes opening up for a glance into the open tomb of otherness. "The memorable is what one can dream about a place" (De Certeau [1980] 1990, p. 163). During our project we encountered dreams, lost ones, unfulfilled ones and hopeful ones. In the encounter with otherness in the "palimpsest place" (De Certeau [1980] 1990, p. 163) of urban life, there seems to be a dynamic of emptiness and fullness, of visibility and invisibility. To clarify this, I will turn to one particular story, one fragment showing how exposure is a work of longing, not of taking possession, and how it opens up new journeys with regards to open spaces, multilingualism and ownership.

Jessie (not her real name) was well known in the neighbourhood as a gender fluid person of colour, identifying as 'she'. After she had come to the Netherlands from a former colony in 1968, Jessie worked as a social worker. To understand her clients better, she spent many holidays in Northern Africa and learned Arabic. Jessie was active in the squatter's movement, until it became more and more difficult to separate her work for the City Social Services from what happened in the neighbourhood: people she knew were her clients and some threatened her to give them what they needed. Jessie saw the neighbourhood changing in the 1990s, which she experienced as an improvement: the families came back and there was more stability. But after COVID-19, she also encountered people who lost their jobs and were indebted. In the congregation, she was proud to be one of our deacons but regretted there were few people with her cultural background. At first, she was disturbed by "hobos and alcoholics" in the church, but she found herself agreeing with what the minister said: "The House of God is open for everybody".[17]

By the time we asked Jessie for an interview for *(T)huis in de Staats,* she was retired and struggled with health problems. We were eager to interview her as someone who knew the neighbourhood very well. It was not easy to get her to share her story; she had to go to the hospital regularly and often cancelled appointments. One of us managed to develop a good relationship with her because of a common Latin-American background. Jessie liked her interview and her picture, taken at her favourite spot on one of the bridges in the neighbourhood. But then, it all went wrong.

When we asked her to be on the cover of our magazine (Dorssers and Meijers 2021), hidden divides in the neighbourhood became visible as well as the tense intersection of belonging to the neighbourhood with other forms of belonging, like family relations. A Muslim woman refused as well, while the white woman who now features on the cover said yes after only one email (but she did not visit the exhibit or participate in the activities). Jessie not only declined our request, but completely retreated from the project. When, due to a misunderstanding, nobody showed up to show her portraits after a Sunday service, she exploded in anger at the Spanish-speaking person she had befriended in the months before. As she refused to have the interview and her picture published, with much regret we took them out of the magazine. Somebody from our group talked to her a few times and she did not go as far as breaking away from the church, something we had been afraid of. But she came only to Sunday service under the condition that nobody would mention what had happened. We respected this.

Unfortunately, very soon after this, her illness got worse and she died. It then became known that she did not want her funeral to take place in the Nassau Church, which bewildered us all. After much deliberation with the family, the memorial service took place elsewhere, but with the participation of our (temporary) minister and other members of the congregation. The service revealed that her family still referred to her as a man, and pictures of Jessie as a young boy were shown. It was clear the family struggled with Jessie's change of gender. But it was agreed that everybody could refer to Jessie the way they knew her. Differences were respected, although we wondered how Jessie would have experienced this. The service was moving and beautiful (Figure 7).

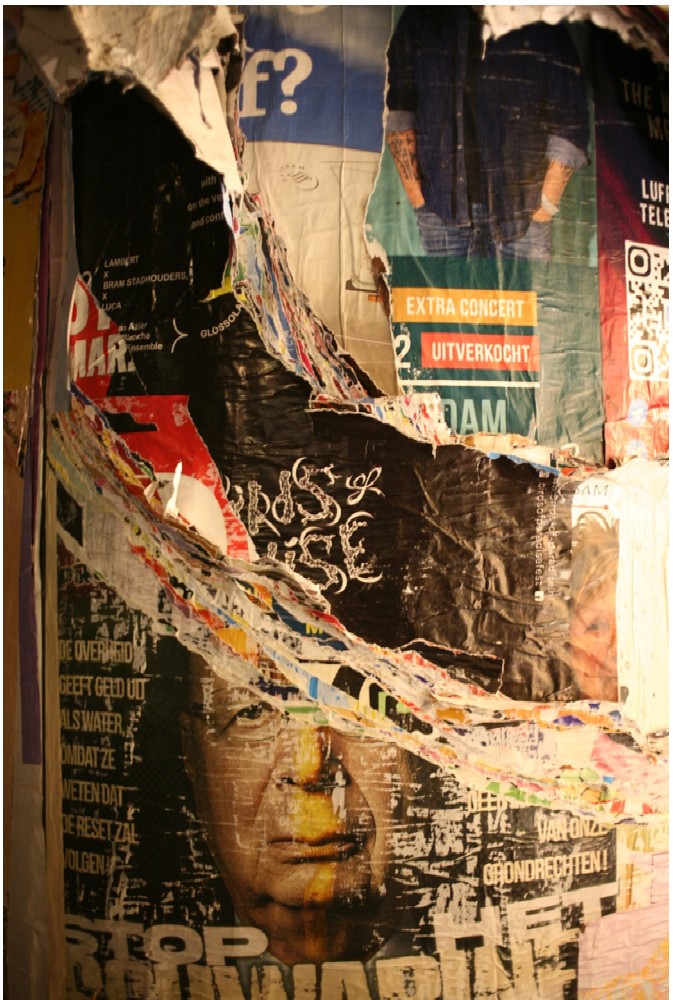

**Figure 7.** Credits: Philippe Velez McIntyre 2023.

Nevertheless, we were left behind with a lot of questions: was her gender one of the reasons Jessie did not want to be on the cover? Had she been afraid of being exposed, maybe especially to the congregation? She had never talked about her gender situation in the Church, and many did not know about it. Did she fear the consequences of such an involuntary coming out, even though her gender was a public secret in the neighbourhood? Had the fear, maybe also in the face of her serious illness, made her explode in anger and withdraw from the project? We had no answers and did not want to fill it in for her now that she could not talk back anymore. Many people, including Jessie, had said they felt at home in the neighbourhood. And although this was probably true, we also realized that underneath this surface of belonging, there were also cultural, religious and gender divides in the neighbourhood. There was racial and sexual discrimination, only visible for those who were confronted with it. Did we do enough to make this visible? The story as she had told us, even unpublished, had been very important in the process to get to know her better. Because she had approved it, we decided to share it with her family, to whom it was an important testimony, and for some, the start for a new approach to Jessie. We are still in contact with one member of her family.

The story of Jessie confronted us with difficult questions about our intentions and the way we had acted. We hoped to "articulate the knowledge that has become silent" (De Certeau [1980] 1990, pp. 162–63). But did we succeed? What is more: were we in the position to do this, as relatively privileged inhabitants of the neighbourhood? In our eagerness to hear Jessie's story, we had not been aware enough of the vulnerability of her position. We had wanted to quickly show her the pictures, but we had hurt her deeply by

not showing up. We acted from a position of power, resulting in a silence from Jessie's side which turned out to be irreparable. Paradoxically, the most important lesson we learned (again) during *T(h)uis in the Staats* might be that exposure is a vulnerable process which demands constant reflection on one's own position and power, both positively and negatively. The open space, the silence in between people and the otherness of the open tomb often are not comfortable places. They are upsetting and disturbing, painful and confronting. It is within this turbulence of grief and regret that our desire is born again. That we grasp for the mercy of the Coming One.

During the project *A(t) home in the Staats*, different groups of people had become visible to each other, like homeless people, those who were well-to-do and the women of the Muslim women organization. But a lot of pain remained invisible. And a lot of knowledge remained underground: secrets, which do not want to be revealed in the public domain, and tactics of ownership not compatible with the openness of our project. What were Jessies' tactics of adaptation and subversion? Her role as a deacon in the Church gave her a public place, but part of her story remained hidden. Even though the sharing of her story after she died brought some relief to her family, we were confronted with the question in what way she had 'owned' her story. Instead of strengthening her ownership, we might have weakened it. We can only do some kind of justice to Jessie's story if we accept that we do not know. There are many perspectives, but hers is not available to us. In order not to kill the mystery of her life, we have to accept that there are multiple fragments and only sometimes, in brief moments, listeners or witnesses. She remains "indefinitely other" (De Certeau [1980] 1990, pp. 141–42).

**Funding:** This research received no external funding.

**Institutional Review Board Statement:** Not applicable.

**Informed Consent Statement:** Not applicable.

**Data Availability Statement:** Not applicable.

**Conflicts of Interest:** The author declares no conflict of interest.

## Notes

1.   Policy paper: *Inspirerende presentie—een nieuwe rol voor de kerk in de stad. Beleidsplan Nassaukerkgemeente* 2018–2022. (Translated from Dutch by the author).
2.   *De Staat van de Stad Amsterdam X, 2018–2019*, OIS. Available online: https://onderzoek.amsterdam.nl/publicatie/de-staat-van-de-stad-amsterdam-x-2018-2019 (accessed on 11 April 2023).
3.   Jan Hendriks on the blurb of Irik 1995.
4.   I have described this project in detail in Meijers 2023, Belonging to the City.
5.   All citations of De Certeau are translated from French by the author, unless stated differently.
6.   The journal Religions uses 'Figures' instead of 'Pictures'. This is not the choice of the author, nor the artist. More on the work of Philippe Velez McIntyre: http://www.philippemcintyre.com (accessed on 13 July 2023).
7.   Comment of Paula Irik to an earlier version of this text, in an email to the author on 27 June 2023.
8.   All citations in Dutch are translated by the author.
9.   This is not a reference. The journal requires that references are made in the text to make it possible to add images to the text. They are put in randomly. The images constitute an independent contribution on the theme 'Walking in the City'. They are not attached to specific parts of the text. This goes for all the references to the images ('figures') in this paper.
10.   This center "supports and trains professionals and volunteers in experienced based learning, who support people living in circumstances in which their humanity is under pressure". Available online: https://korschippers.nl/overons/ (accessed on 13 April 2023)
11.   Comment of the chairman of the Church Council of the Nassau church, in an email to the author, 25 June 2023.
12.   Reports of exposurewalks in 2018 by members of the Nassau church (Archive exposure group).
13.   For the English translation of this essay, I use 'Walking in the City'. Available online: https://www.mobilistiek.nl/assets/Uploads/Downloads/Michel-de-Certeau-Walking-in-the-City.pdf (accessed on 24 April 2022).

[14]　　Many thanks to Jaap Buning, Paula Irik, Dick Jansen, Annelies Faber, Philippe Velez McIntyre, Luc Tanja, Gerrie Willemsen for critically commenting on an earlier version of this text and bringing in their experience of our journey.

[15]　　See for example the art project of Ine Gevers in 2016, *Hacking Habitat*, which meant to take back control in a world controlled by technology. https://hackinghabitat.com/nl/over-hacking-habitat/ (accessed on 13 June 2023). Cf. (Meijers 2015b).

[16]　　Comment of the chairman of the Church Council of the Nassau church, in an email to the author, 25 June 2023. The topic was discussed in the church council on 5 July 2021.

[17]　　Unpublished interview, archive *A(t) home in the Staats*.

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
