# Peer review of "Walking in the City: Christian Spirituality in Amsterdam through the Eyes of Michel de Certeau"

_religions, doi:10.3390/rel14080968_

Round 1

Reviewer 1 Report

I found this one of the best urban theological contributions I encountered in recent times. The search for God beyond the empty tomb, the question of Mary Magdalene, almost desperately crying out "Where are you?", fused with insight into urban spatiality, urban change and the effects of gentrification, and further fused with profound narratives of people and spaces in the vicinity of the Nassau Church, gives for a very compelling perspective on urban Christian spirituality. The metaphors used in the 6 headings provide beautiful lenses with which to approach changing urban contexts, contemporary and practical. The writing style is such that one feels as if one is walking the streets around the church, the bridges crossing canals, and encountering the people related in the article. I do not suggest any improvements really because I was moved by this contribution. It combines scholarly merit, embodied immersion and a spirituality of the empty tomb. 

Author Response

Dear reviewer,

thank you very much for your kind review. I was hoping that my scholarly insights and personal experiences would together be able to contribute to issues of being church in the city and I am very glad you have experienced it as such. In the meantime, I received comments of several readers engaged in the Nassauchurch and the neighbourhood, and I will use them to improve the paper.

Reviewer 2 Report

Overall I enjoyed reading your article. I think you handled DeCerteau well and his work fit well with your overall work.  

I have a few suggestions for you:

The article is co-authored but then starts with “I will.” I suggest consistent voice throughout. If it’s co-authored then “we” should be used.  

Edits: 
Line 54: it’s should be its

Line 57: Should it be economic change?
Lines 304-308 are a bit cumbersome. Suggest edit.

Lines 488-491 rewrite in more professional tone

I made a few edits which I listed above. 

Author Response

Dear reviewer,

thank you for reading my paper and commenting on it. I will be happy to improve it thanks to your comments. The article is not co-authored though, I am the only author. Others have read and commented on it, people who know the Nassauchurch and the neighbourhood very well. The project I mention was done by a group, of which I was a member. But the article was only written by me. I will make sure this is more clear.

I will also critically look at your other comments. Thank you!